# Undervalued essential work and lacking health literacy as determinants of COVID-19 infection risks: a qualitative interview study among foreign-born workers in Sweden

Mia Söderberg [iD],[1] Maria Magnusson,[2] Juhaina Swaid,[1] Kristina Jakobsson,[3] Annika Rosengren[4]

For numbered affiliations see end of article.

**Correspondence to**
Mia Söderberg;
mia.soderberg@amm.gu.se

## ABSTRACT

**Objectives** To investigate work and living conditions as determinants of COVID-19 infection risks in foreign-born workers in non-healthcare occupations.

**Design** Data were collected according to a qualitative design, using semistructured interviews. Verbatim transcripts of these interviews were analysed according to systematic text condensation.

**Participants** We recruited foreign-born workers (n=15) and union representatives (n=6) among taxi drivers, bus and tram drivers, pizza bakers, cleaners and property caretakers, all indicated as risk occupations during COVID-19 in Sweden.

**Results** Four overarching themes were found: 'virus exposure at work', 'aspects of low status and undervalued work', 'lack of access to information' and 'foreign-born persons' position'. Virus exposure was frequent due to many social interactions over a workday, out of which several were physically close, sometimes to the point of touching. The respondents fulfilled important societal functions, but their work was undervalued due to low job status, and they had little influence on improving safety at work. Lack of health literacy limited foreign-born workers to access information about COVID-19 infection risks and protection, since most information from health organisations and employers was only available in Swedish and not adapted to their living conditions or disseminated through unknown channels. Instead, many turned to personal contacts or social media, through which a lot of misinformation was spread. Foreign-born persons were also subjected to exploitation since a Swedish residency permit could depend on maintaining employment, making it almost impossible to make demands for improved safety at work.

**Conclusions** Structural factors and a lack of adapted information manifested themselves as fewer possibilities for protection against COVID-19. In a globalised world, new widespread diseases are likely to occur, and more knowledge is needed to protect all workers equally. Our results are transferable to similar contexts and bring forth aspects that can be tried in quantitative studies or public health interventions.Cite Now

## STRENGTHS AND LIMITATIONS OF THIS STUDY

⇒ This study recruited participants from an already vulnerable sociodemographic group who needed tailored public health interventions during the pandemic. However, since persons from this subpopulation tend to not participate in scientific studies, little was known about their particular needs.

⇒ The use of qualitative methodology increased the possibilities of exploring new themes and extracting in-depth information about such themes.

⇒ The interviews were mostly conducted in the participant's native language, but some interviews were carried out in a language that was neither the respondent's nor the interviewer's native language, which likely restricted a deeper understanding.

⇒ Those who volunteered may have consisted of a selection of more well-established persons in Swedish society. Moreover, almost all the participants were men.

## BACKGROUND

SARS-CoV-2 is an RNA virus that causes COVID-19, which had a pandemic spread over the winter and spring of 2020, reflected in high morbidity and mortality worldwide.[1-3] When the WHO declared the outbreak as a pandemic (11 March 2020), the Swedish Public Health Agency made formal requests to impose social restrictions, such as a maximum number of people at social gatherings, that elderly people (>70 years) should avoid close social contacts, that there should be no outside visitors at elderly homes and that the teaching of upper secondary school pupils and university students should be done remotely.[4] Otherwise, Swedish society remained open, with no curfews, and with shops, restaurants and elementary schools mostly operating as usual. In September 2020, when the prevalence of COVID-19 cases

was low, most recommendations were relaxed, but reintroduced again in December 2020, when the numbers of cases increased rapidly. This meant stricter regulations for the allowed number of people in shops and social venues, earlier closing times for bars and restaurants, and recommendations to wear facemasks when travelling with public transport, although the latter was not mandatory.

In December 2020, the Swedish COVID-19 vaccination programme started. The first batches were aimed at health and social care staff, the elderly or those who were considered as risk groups due to a diagnosis such as cardiovascular diseases or diabetes. In June 2021, all adults ≥18 years were eligible for vaccination, which was expanded to everyone aged 16 years or older in July of the same year. The vaccination programme was deemed successful as over 70% of the adult Swedish population has received at least one dose as of July 2021.[4] As a result, almost all regulations were removed but were once again reinforced in December 2021 when the number of confirmed cases increased. Finally, most recommendations were removed in February 2022 and in March 2022, when COVID-19 was no longer considered a public and societal dangerous disease, the Swedish Public Health Agency directed their main focus on increasing vaccination rates and vaccination booster jabs.[4]

Due to the absence of any treatment or vaccine during the first waves of the outbreak, research was foremost concentrated on clinical and pharmaceutical aspects. Yet, other public health concerns emerged as markers of a low socioeconomic position or belonging to an ethnic minority were associated with an increased risk of infection, severe illness and death from COVID-19, increasing already existing health disparities between social groups.[5][6] Severe disease and death are associated with the prevalence of underlying diseases.[7] Infection risk, however, is determined by the number of social contacts as COVID-19 is primarily transmitted through drops in sneezing, coughing, speech or aerosol transmission, with the latter being exacerbated in crowded and poorly ventilated spaces. In people of working age, work is a dominant source of social contacts. However, there are markedly differences in possibilities to avoid virus exposure, as low-wage workers to a less extent can practise social distancing from the nature of their work tasks and to work from home, and reliance on public transport when lacking the economical means to own a car.[8][9]

When COVID-19 was declared a pandemic, work in healthcare and social care was promptly identified as a risk for becoming infected due to many close physical contacts. Yet, there were other occupations with many social contacts in which a high prevalence of COVID-19 infection was observed, and in which virus exposure and protection possibilities were less investigated. In a review study, including studies from the early stages of the pandemic, abattoir workers and related food workers, and retail staff displayed increased risks of contracting the coronavirus.[10] Two Swedish register studies, investigating occupations outside the health and social care sector,

showed the highest prevalence of a positive PCR test for COVID-19 or death in COVID-19 (until May or June 2020) among taxi drivers, pizza bakers, and bus and tram drivers and a high prevalence among property caretakers and cleaners.[11][12] Another study, recording COVID-19 cases over the first 40 days since the first detected case in six Asian countries, noted the highest infection rates in taxi and van drivers and, in descending order, shop salespersons, religious professionals and construction labourers.[13]

As time progressed, similar and new at-risk occupations were found. A study from the UK, with access to national register data until December 2020, showed the highest COVID-19 mortality rates in taxi drivers (HR 2.01, 95% CI 1.67 to 2.43, in men), followed by support staff, bus and coach drivers, sanitary workers and van drivers, adjusted for age, multiple sociodemographic factors and prior diagnoses.[14] In a Norwegian national register study, the highest ORs for COVID-19 (a positive PCR test or an International Classification of Diseases (ICD)-10 diagnostic code U07.1) over winter–spring 2020 were found in bus and tram drivers, taxi drivers, food counter attendants, bartenders, shop assistants and cleaners.[15] During the winter of 2021, which constituted the second pandemic wave, public transport conductors, bartenders, travel stewards, waiters, food counter attendants, bus and tram drivers, taxi drivers and hairdressers displayed the highest COVID-19 infection rates.[15] A report from the Swedish Social Insurance Agency, using sick leave data up until March 2021, illustrated a 42% increased relative risk of long-term sick leave (>21 days) in taxi drivers, compared with all other employed persons.[16] Bus and tram drivers also displayed higher rates of both short-term sick leave (42% increased risk of at least 1 day of sick leave) and long-term sick leave (39% increased risk of >21 sick leave days).

Despite the observations of an increased risk of COVID-19 in several occupations with similar characteristics, little is known about infection risks in day-to-day situations as reported by the workers themselves. An interview study with fast-food workers in the USA (n=15) provided some insight, such as customers refusing to wear facemasks or subjecting workers to verbal abuse or threats when they were being asked to practise social distancing or to wear a facemask.[17] A British study, based on interviews with bus drivers, delivery workers and police staff, illustrated a lack of, or delayed, access to protective equipment and difficulties in distancing themselves from other people.[18]

Higher risks of becoming infected may also interplay with having an immigrant background as most of the Swedish occupations with the highest occurrence of COVID-19 also employ the largest proportion of foreign-born workers: pizza bakers (79%), cleaners (57%), bus and tram drivers (48%) and taxi drivers (47%).[19] Analyses based on Norwegian national register data found that workers from Somalia, Pakistan, Iraq, Afghanistan and Turkey displayed substantially higher ORs of

a COVID-19 diagnosis than the general population, in particular bus and tram drivers (OR 12.72, 95% CI 8.29 to 19.55), cleaners (OR 8.81, 95% CI 6.66 to 11.66) and taxi drivers (OR 6.56, 95% CI 4.88 to 8.86), age and sex adjusted.[20] Yet, it was unclear if this was related to occupational risks or other factors, as foreign-born persons in the listed occupations mostly did not display higher ORs than other immigrants from the same country. Similarly, in a British registry study, the highest mortality from COVID-19 was found among taxi drivers, bus drivers and cleaners, but the risk estimates decreased markedly (taxi drivers and cleaners) or were nullified (bus drivers) when adjusting for age, ethnicity, housing, living in a socioeconomically vulnerable area and relevant diseases.[21]

The reasons for increased infection risks in foreign-born workers are not clear. Besides virus exposure at work, one aspect that has been brought forth is the lack of access to crisis information, as well as information on safety and rights at work.[22 23] Such barriers relate to 'health literacy', a concept that can be briefly explained as the use of information to protect and improve health.[24] From an individual perspective, the basic level involves having adequate language skills to consume and understand information correctly, while more complex features reflect the ability to acquire relevant information and how to apply it. Health literacy also reflects organisations' ability to adapt appropriate support for health-promoting behaviour. Another potential risk factor is poor housing conditions, which are more commonly found among foreign-born persons. Sweden displays some of the largest within-country gaps in inequalities in housing deprivation in Europe, as around 30% of all foreign-born persons live in overcrowded households, compared with 9% of Swedish-born persons.[25] Overcrowding contributes to the spread of infection, and if cohabitants in such households work in risk occupations for COVID-19, it can exacerbate a circular infection pattern between the work and home environment.

In summary, increased COVID-19 infection rates have been found in many occupations outside the health and social care sector, and foreign-born workers in these jobs seem to be at particular risk. Based on existing evidence, it can be assumed that increased infection relates to difficulties in social distancing, as well as barriers to accessing crisis information. Yet, exposure risks in the everyday situation as described by foreign-born workers themselves are largely unknown, partly because these occupations have received less attention but also since this sociodemographic group tends to be less prone to participate in scientific studies.[26] This explorative study aims to increase the understanding of pathways for COVID-19 infection in foreign-born workers in non-healthcare occupations. We used a qualitative in-depth design to capture aspects that may not be adequately covered using questionnaires or other methods collecting quantitative data.

## METHODS

### Recruitment of participants

In order to complement prior studies on non-healthcare keyworkers and based on official statistics from the Public Health Agency of Sweden, we chose to investigate the study objectives in pizza bakers, taxi drivers, bus and tram drivers, cleaners and property caretakers.[11 12] The participants were recruited through convenience sampling, a non-random sampling method in which participants are invited through existing social networks. Initially, we only recruited foreign-born workers, but as the project progressed, we decided to include union representatives, who could contribute with an overview of the selected occupations in addition to their own experience of the working conditions. The union representatives did not have to be foreign-born. Two health guides from the East Gothenburg Health Centre were contracted for the recruitment: Juhaina Swaid (JS) (coauthor) and Saido Mohamed (SM). In Sweden, health guides have a broad range of tasks such as facilitating communication between organisations and citizens, disseminating knowledge about how people can promote their health and navigating the healthcare system to ensure that everyone in society has access to the same information. Through their work, JS and SM had a wide network of social contacts and an established trust with the people in their catchment areas. They could also conduct the interviews in several of the participants' native languages.

The participants in this study were recruited through (1) social contacts of the health guides JS and SM (n=13), and (2) social contacts of colleagues at the Department of Occupational and Environmental Medicine, Sahlgrenska University Hospital (n=2). The union representatives (n=6) were recruited by (3) emailing the unions' contact persons listed on the website for each union. When contact was established, the person in question sent an invitation email to either a selection of union workers who might be interested in participating or to a global email list. In Sweden, most pizza bakers are self-employed or work in small family-owned businesses, and we were unable to find any union representatives for this occupation. The lack of union representatives for this restaurant workers' branch was confirmed by the Swedish Hotel and Restaurant Workers' Union.

### Patient and public involvement

No patients were involved in this study.

### Data collection

The workers were interviewed during March 2021–March 2022, and the union representatives were interviewed in February–March 2022. To encourage the participants to freely describe their situation, the interviews were carried out according to an explorative and semistructured methodology with sets of both open-ended questions and specific questions about the work environment. The interviews were carried out by first author Mia Söderberg (MS), and the two health guides JS and SM. MS

had previous experience in collecting data according to qualitative methodology. JS and SM received supervision from MS on how to conduct the interviews. JS also participated as an observer during one of the interviews that MS conducted, but this was not possible for SM. Most of the interviews took place at the participant's workplace or by Zoom or telephone. A few of the interviews were carried out at the homes of the participants or a café in their neighbourhood. The reason for the different venues was to accommodate the wishes of the participants, out of which several were hesitant to participate, or worked long hours and had many responsibilities at home, and therefore requested to be interviewed by phone or Zoom, or near their homes in an environment where they felt comfortable.

The interview guide was formulated around three main topics: (1) virus exposure and protective measures at work, (2) access to information and (3) living conditions. Examples of the interview questions are as follows: (virus exposure and protective measures) 'How often do social contacts at work require other people to stand very close to you and shout to overcome the surrounding noise?', 'What methods have you used to protect yourself from exposure to infection?'; (access to information) 'How do you obtain information about COVID-19?'; (living conditions) 'Are there any reasons why you would not call in sick if you had COVID-19 symptoms?'. We also included an open question: 'What do you think is the main reason for the spread of COVID-19?'. The union representatives were asked the same questions both from the perspective of their own work conditions and the perspective of the members they represented. Both JS and SM had encountered some of the workers prior to this study through their work as health guides. For that reason, the interviewers took particular care to explain that the interviews and handling of any sensitive information were conducted according to professional secrecy. None of the other authors had prior contact with the participants. After the 15th interview with the workers, the content became repetitive and the data were perceived as saturated; therefore, the recruitment ended. All union representatives who reported an interest in participating were interviewed.

### Language considerations and transcription

The interviews were carried out by JS (native language Arabic), SM (native language Somali) and MS (native language Swedish). The languages of each interviewer are shown in figure 1.

None of the 15 foreign-born workers had Swedish as their native language. Five of these interviews were conducted in Arabic by JS and four interviews in Somali by SM, that is, an interview situation in the native language of both persons. The remaining six workers were interviewed in Swedish, of which four were conducted by JS, that is, in a language that was neither the interviewer's nor the respondent's native language. With this in mind, JS revised the transcript of these interviews to evaluate if it corresponded with her perception of what emerged during the interviews and to clarify aspects that may not be apparent. The six union representatives were interviewed in Swedish by MS. Swedish was not the native language of three of the union representatives, but their language level was perceived as either equal to a native Swede (native language: Norwegian) or as an advanced Swedish speaker (native languages: Serbian and Persian). The interviews in Arabic and Somali were translated into Swedish by professional translators, and all 21 interviews were then transcribed verbatim by the same company that translated the interviews. After this process, JS and SM contacted a few participants again to get further information on unclear content.

### Analyses

The analysis of the data was primarily carried out by first author MS and second author Maria Magnusson (MM), who were both experienced in carrying out qualitative studies. MM has particular experience with systematic text condensation as proposed by Malterud,[27] a descriptive and explorative method, which was the chosen analysis method of this study. This methodology is also characterised by paying particular attention to the participants' experiences and perspectives while emphasising an understanding of the underlying processes. MS and MM started by reading the entire transcript of all the interviews, without singling out meaningful information to get the overall gist of the material. Both authors then

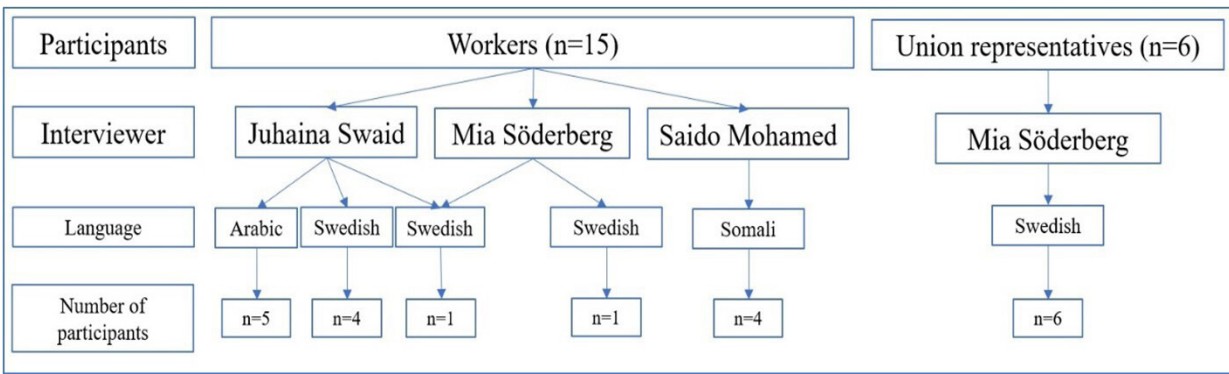

**Figure 1** Languages used during the interviews.

reread the printed transcripts and marked all content that was perceived as meaning-bearing units with a focus on the study objectives. Other content was marked to be re-evaluated later.

In the next step, the marked units were revised and labelled into either relating to work, general living conditions, information access or aspects of being foreign born. From this step onwards, JS participated in the analysis process with interpretations and her professional and cultural knowledge. The themes were then further refined and presented for discussion among all the authors. The senior researchers, Kristina Jakobsson (KJ) and Annika Rosengren (AR), contributed with their expertise in public health and occupational research. The result was then edited and condensed into main themes with underlying subthemes. The subthemes were also evaluated with consideration to how often a certain theme was mentioned, or if the union representatives reported this as a common occurrence among the members they represented.

## RESULTS

Fifteen workers aged 30–50 years were interviewed, and a majority were men (n=13). The workers were pizza bakers (n=4), property caretakers (n=2), taxi drivers (n=3), tram or bus drivers (n=3), and cleaners (n=3), out of which two worked as hospital cleaners. Time in their current occupation ranged from 2 to 16 years. Six union representatives participated, three men and three women, representing property caretakers (n=1), cleaners (n=3), taxi drivers (n=1) and bus drivers (n=1). The union representatives worked with union assignments 1–4 days per week, the remaining time they worked in their main occupation. The number of unionised members they represented ranged from 300 to 7000 members. The countries of origin of the participants were Turkey, Syria, Iraq, Palestine, Serbia, Somalia, Norway, Iran and Sweden. All the Swedish-born participants were union representatives. Most of the participants lived together with their spouses and children. Further descriptive information about the study participants can be found in online supplemental appendix 1.

Our analyses resulted in four overarching themes: (1) virus exposure at work, (2) aspects of low status and undervalued work, (3) lack of access to information and (4) foreign-born persons' position. The main themes and subthemes are presented in table 1.

### Virus exposure at work

Due to the nature of the work tasks and the organisational environment, the participants reported that it was

**Table 1** Main themes and subthemes

| Main themes | Subthemes | |
|---|---|---|
| 1. Virus exposure at work | 1.1 Hard-to-avoid contamination risks | Social contacts<br>Surfaces |
| | 1.2 Lack of safety culture | Employees<br>Employers<br>Employer liability |
| | 1.3 Contributing to the spread of the disease | Social contacts over many venues<br>Social contact with vulnerable persons |
| 2. Aspects of low status and undervalued work | 2.1 Fulfilling an unappreciated societal role | Increased need for cleaning during the pandemic<br>Upholding the public transport system |
| | 2.2 Occupational status influences workers' value and safety | Protective interventions deemed too expensive<br>Unequal protection |
| | 2.3 Low authority over their work environment | Lack of influence<br>Distance to decision-makers |
| 3. Lack of access to information | 3.1 Inadequate health literacy | Workers<br>Employers<br>Organisations |
| | 3.2 Consequences | Turned to unofficial information sources<br>Spread of misinformation |
| | 3.3 Suggestions for improvements | Information in many languages<br>Using complementary information outlets<br>Audio-based information |
| 4. Foreign-born persons' position | 4.1 Exploitation of a vulnerable position | Need the job for a Swedish residency<br>Need to financially support others<br>Unawareness of regulations or rights |
| | 4.2 Protective directives not adapted for persons outside the majority of society | Cannot work from home<br>Cannot avoid public transport<br>Different social patterns |

nearly impossible to avoid being exposed to infection risks. Subthemes for 'virus exposure at work' were sorted into 'hard-to-avoid contamination risks', 'lack of safety culture' and 'contributing to the spread of disease'.

### Hard-to-avoid contamination risks

The most notable risk of being exposed to infection appeared to be the many social contacts the participants encountered over a workday, out of which several were physically close, and could include touching other people. For example, some bus, tram or taxi companies did not reinstall any protective screens between the driver area and passengers during the second wave (winter 2021) and demanded that passengers enter through the front doors to facilitate ticket inspection. The drivers estimated that they would then come in close contact with more than hundreds of people in a single workday. The bus, tram and taxi drivers also carried out mobility services for citizens with reduced functional ability, such as the elderly, physically disabled persons or schoolchildren with special needs, who often needed hands-on support.

> Bus driver: For me, the passenger is the cause of infection. I drive you to work, to your exercise, to the store, to the health centre. One guy who stood and talked to me last week (comment: this interview was conducted in February 2021). He was going to the health centre for a COVID test! What the h*ll are you standing here saying, I thought.

> Taxi driver: There are some who are ill. I help elderly people who can't sit down, put on their seat belts, or walk up the stairs.

> Interviewer: Do you support them?

> Taxi driver: Maybe I'll hold their hand, yes.

> Taxi driver: Many of them need help with the seat belt and maybe help with getting into the car…(if) they have a walker - you go out and meet them, if it's slippery outside, you have to hold their arm… then you open the door and you want to hold their head so that they don't hit their head on the ceiling… yes, we touch a lot of people, even if they can manage quite well on paper. But you can't let anything happen. Say we touch 50% of all adults. I would say 80% need help with the seatbelt.

The participants also encountered many social interactions that were not part of their work tasks, foremost in crowded venues such as schools and shopping centres, and the hospital cleaners came into contact with patients and medical staff. Many cleaners also travelled by public transport between their daily assignments since they did not own a car, or to avoid parking fees and traffic jams, which exposed them to risks, especially since facemasks were not mandatory on public transport in Sweden. The bus and tram drivers would also use public transport to travel to the start and end stations of their shifts.

> Hospital cleaner: The patients need help. We say no! This is not part of our job. But if they are old and can't find the buttons (comment: the buttons used to call for help or to operate their bed), then we will show them the buttons. Some of the patients are old, then you have to stand close and speak loudly.

> Hospital cleaner: Always when I enter this office, the doctor is sitting there. He doesn't want to leave. So, I wipe a bit there (comment: the participant cleans the area around where the doctor is sitting).

> Union representative for cleaners: I have a car, but I don't want to drive. You end up in traffic jams and you can't find a parking space. There's a lot of time pressure in our work… Therefore, people are forced to go for example by subway, by bus. I can see that the people who use public transport do not use facemasks.

The participants handled tools or surfaces that had been touched by many other people, especially the property caretakers and the cleaners who operated over large areas, cleaned many different surfaces and handled waste. The hospital cleaners rarely had the time to sanitise their equipment despite serving hospital wards, waiting rooms and toilets.

> Hospital cleaner: In our group, we are five staff, and we use the same materials every day.

> Interviewer: How often is it cleaned?

> Hospital cleaner: We usually take care of our materials and stuff and machines once a week.

> Interviewer: You all use the same materials and the same machines?

> Hospital cleaner: Yes.

> Interviewer: The machines and tools you use, have you increased cleaning them?

> Hospital cleaner: No.

### Lack of safety culture

The workers made an effort to work safely and made many requests for protective equipment at the beginning of the pandemic, but also reported that this became less important as time passed, due to stress, feeling restrained by cumbersome protective equipment or reduced fears of the disease.

> Union representative for cleaners: In the beginning, it was a panic about protective equipment, that's when everyone wanted to have and have and have, and finally when we got enough cleaning materials and everything they asked for that's when people started to ignore it. People eventually got used to it and said, 'But nothing happens.'

Some employers introduced safety procedures and provided their workers with personal protective equipment in the early phases of the pandemic, while others seemingly delayed the provision of protective equipment

and only did so after being pressured. The participants speculated that this was related to employers' lack of knowledge of best practices, not enough resources or that they were unmotivated to protect their staff in a 'soon-to-be-over' crisis.

> Hospital cleaner: We got facemasks recently. The last time when corona was getting worse… they got new rules for us that we have to use facemasks (comment: they received facemasks in October 2020).

> Bus driver: No, we didn't receive any recommendation from April to May 2020. No one warned us about this. It came after the number of infected people increased. Then the company and the municipalities began to take action and give recommendations. They started with it after the month of July (comment: July 2020).

> Hospital cleaner: In the beginning, they sent us to a department called ICU [intensive care unit] and it was full of patients who had had COVID. We went there without protection, without mouth protection (comment: the unit was cleared of patients before cleaning). There are so many instructions, and we know all the instructions but in the workplace, it is the opposite. It is the opposite.

> Hospital cleaner: The emergency department built a barrack for examinations. Those who came were unwell. They had Corona or fever, or something. We asked why we are cleaning there - it's a risk! He (comment: their manager) said you must do it, there is no risk.

There was also a feeling that employers only made changes that looked good 'on paper' but neither cared for and sometimes even contradicted safety protocols. For example, managers ignored when staff came to work with obvious COVID-19 symptoms or that lunch breaks were scheduled so that far more than eight people would be gathered in a small lunchroom that was not properly cleaned or sanitised. People who spoke up could be punished by being assigned 'the graveyard shifts' or not receiving enough work to fill a full-time schedule.

> Hospital cleaner: They say we can't have staff meetings because of being too many. When we're in the lunchroom there are more than eight people there… But I have to sit and eat!… The boss, we think he should make (changes). He doesn't! (comment: eight people was the maximum allowed number of people at social gatherings at the time of the interview)

Unclear employer liability also made it difficult to demand safer working conditions. For example, mobility services carried out by taxi drivers were often purchased by the local borough, while one company owned the vehicle, and another company might employ the driver. There was also uncertainty regarding who was responsible for the protection of cleaners against social contacts at shopping centres or in schools.

> Union representative for bus drivers: There are several responsible persons… the one who owns the car, the company we work for, and then the one who procured the travel service - therefore there is no one to report problems to… I got frustrated and emailed the travel services committee (comment: a committee of local politicians), one, two, three different persons, and got no answers.

### Contributing to the spread of the disease

Some participants worried that their work contributed to the spread of COVID-19 to other, sometimes vulnerable, people. When conducting mobility services, the taxi drivers drove up to four passengers who were sometimes picked up from crowded venues, such as community centres or schools. This caused concern as many passengers could be considered at-risk groups for severe COVID-19 due to old age or poor health. The hospital cleaners met both healthcare staff and patients, which meant there was a large risk of contracting and spreading the virus.

> Union representative for taxi drivers: This is quite a large day centre… say there are 20 people, plus regular staff… and there are maybe 5–6 other residents who go to other daily activities and there are staff so if you think of the infection… I pick up this girl and then I have to pick up a regular service traveller and there is an elderly man with a facemask, and I have to put him in the back seat with her (comment: the girl that was picked up from the service centre) and that didn't feel good.

> Interviewer: Did she have a facemask?

> Union representative for taxi drivers: No no, no possibility of wearing a facemask no… it was so offensive against everyone involved.

### Aspects of low status and undervalued work

Despite providing important societal services, there was a feeling that because they worked in what was considered low-status job, their safety and the work they performed were not considered important and valued. These aspects were subcategorised into 'fulfilling an unappreciated societal role', 'occupational status influences workers' value and safety' and 'low authority over their work environment'.

### Fulfilling an unappreciated societal role

Many participants had expanded responsibilities during the pandemic and were proud of upholding important functions and contributing to the battle against COVID-19. They received appreciation for their hard work but worried that this would be forgotten once the pandemic was over. There was also anger that contractors demanded extra work but did not want to offer any additional pay.

> Union representative for bus drivers: We feel very forgotten in this. If we stop all traffic, then half of

Sweden stops because then the health care workers can't reach their destination, then the teachers and students can't reach their destination. We are equal, we are not health care staff, but we are important in society.

Union representative for cleaners: I think we are an important professional class, but in society, it is classified as a low professional class… We got a lot of extra orders, and they were very positive, and they were grateful for the cleaning and thanked all the cleaning companies in the community. But guaranteed, they will forget everything when the pandemic is over, unfortunately.

### Occupational status influences workers' value and safety

Many companies seemed to have treated the workers' safety as a balancing act between protecting their staff and a loss of revenue. Some transport companies did not want to install protective screens or restrict the number of allowed passengers per taxi car during the second wave (winter 2021), since that would require extra cars and staff. There were also bus companies that resisted introducing facemasks to not 'scare the passengers'. When facemasks or visors became mandatory, this posed new problems as wearing a visor would fog up glasses or could cause mirror reflections, endangering driving. Some buses received a driving ban due to these mirror reflections but were put into service anyway. The union demanded restrictions from passengers entering the area near the driver, but this was met by resistance since it would make ticket inspections impossible, leading to losses in revenue.

Union representative for cleaners: When the first wave came, I contacted some employers to make checklists with them regarding infection (risks), they would then say, "But they already have gloves!"

Union representative for bus drivers: They lost 1.6 billion (Swedish krona) last year. If you divide 1.6 billion by the number of drivers, you know the price of my life. Someone who sits in an office decides they want to gain money in this way (comment: a loss equal to around US$160 400 000).

Union representative for taxi drivers: Then the infection increased and it's like, well… the front seat was closed again at the beginning of January. So, during this whole giant wave, it was open… it was awful… I called my boss and said, "I won't put anyone in the front seat." He completely panicked and said, "You can't do that," well yes actually I can because it's about my life and health (comment: confirmed COVID-19 infection cases increased rapidly in November 2021, and the front seat was closed for passengers January 2022).

The hospital cleaners were initially refused antigen tests at work, despite that this was freely available for the healthcare staff working on the same premises. Instead, the cleaners had to go to their local healthcare centre to get tested and were not allowed to work while waiting for the results, which meant a loss of income.

Hospital cleaner: There is a special test at the hospital for staff there. But they sent us to our health care centre… We have to make an appointment ourselves and stay at home for four days because of this until we get an answer. Now I think it will be better… but this thing started two months ago (comment: free antigen tests were available to the cleaners at their workplace in February 2021).

### Low authority over their work environment

Most of the workers had limited possibilities to influence their work conditions or to make their demands and concerns heard by the management or the employers. The people who made the decisions on safety rules and other work-related aspects were far removed from the day-to-day work and had limited knowledge of the actual work content. There was also anger that those in charge worked in a safe environment from home, did not listen to workers or union representatives, and seemingly looked down on the workers.

Union representative for taxi drivers: It's about us being just taxi drivers. Yes, and that's the hierarchy, yes… and it's really sad to say it, but that mentality is out there (comment: among the decision makers), they are dumb taxi drivers, it's impossible to get away from that.

Union representative for bus drivers: You sit at home, in your safe quiet living room while I'm out there driving among people and (you) talk about how I should feel at my workplace?! Damn it! Do not sit at home in your living room and think you know better than me about my workday.

### Lack of access to information

Throughout the interviews, it was clear that faltering health literacy resulted in a lack of access to vital information regarding infection risks and protective measures. This theme was sorted into the subthemes 'inadequate health literacy', 'consequences' and 'suggestions for improvements'.

### Inadequate health literacy

Almost all participants reported that they, their colleagues or some of the union-affiliated members they represented had insufficient language skills to properly understand crisis information in Swedish, especially if such information was formulated in a bureaucratic style or with ambiguous phrasing. Due to a lack of information about rights even before the pandemic, many were also unsure what demand they could make regarding safety at work and provision of protective equipment when COVID-19 spread through society.

Taxi driver: The news reporting about the disease was in Swedish, there was no Arabic channel that gave information about it in Arabic or other languages. There are many people in Sweden who do not know Swedish. They know limited Swedish, which is not good enough.

Pizza baker: What I don't understand is – there's no requirement for facemasks. And then they say that we should wear facemasks when it is… high time? Or when it is rush hour? I cannot know what time it is rush hour… should I wear a facemask or not?

Union representative for cleaners: As I said, we have a bit of a problem with the language. People might feel worried if they want to complain or report something, they don't know that they have the right to do so. They get scared and don't dare say anything. That's the lack we actually have.

Health literacy is a two-sided process and involves the competency of the information disseminator, but as of winter 2022, many employers still only distribute safety information in Swedish and through ineffective methods. Taxi, bus and tram drivers and property caretakers, whose routes and assignments need to be coordinated, received information via staff apps or web platforms. For most others, information was spread through methods such as text messages or leaflets on the staff noticeboards. This information was mostly in Swedish, and the participants figured that their employers knew that some workers would not understand, but that it 'looked good on paper'. Some participants received oral information at staff meetings, but this was also in Swedish.

Union representative for cleaners: They have an information board in the office, people go by and collect materials or gather there. But they just put information on the board, I think you should say that this information applies to Corona so go and read it for example.

Interviewer: They think they have done their employer responsibility, but they don't check that people actually read and if they understand?

Union representative for cleaners: No exactly.

Bus driver: The Swedes are a people who trust the government, (but) the Swedish media have not explained clearly, in a good way about corona for migrants or new arrivals, the consequence of this was that a large part of them died from corona.

Hospital cleaner: In our job, there are many foreigners. They don't understand the language very well. Sometimes they sit and listen (comment: during the staff meetings) but they don't understand anything.

## Consequences

The lack of adapted information from employers and official outlets resulted in almost none of the workers using official public health institutions as a source of information, at least not during the first stages of the pandemic. For many of them, the Public Health Agency of Sweden or the healthcare information organisation '1177', the main distributors of COVID-19-related information, was unknown. The workers primarily received their information from news channels in their country of origin, from friends and relatives, or social media, through which a lot of misinformation was spread. Some participants also mentioned institutions that they were familiar with and trusted, such as their local healthcare centre or pharmacist, as reliable information sources.

Taxi driver: At the beginning, a friend, a doctor, said that corona did not exist. Such information has confused us a bit, hasn't it? A doctor… who says that there is no such thing as corona. In the beginning, I heard many rumours, that there are many old people in Sweden that they want to get rid of, right? In the beginning, we heard a lot of contradictory information.

Interviewer: 1177 have a webpage that is translated into Arabic. If something happens, would you call…?

Property caretaker: I did not make contact with anyone because of what was spread in the media, Facebook, and from the doctors. It was different (conflicting) information. I thought it was not beneficial.

Interviewer: If you want to obtain information about COVID-19, which information source do you trust?

Property caretaker: My relatives.

Interviewer: Which source of information do you trust the most?

Taxi driver: Ah…TV. There are many people who watch TV, and the radio as well (comment: TV and radio from their home countries).

Interviewer: Do you trust the authorities?

Property caretaker: No, I don't. I am newly arrived in Sweden, there are many websites that are linked to the authorities that I am not aware of. I don't know who to contact. I might contact the health centre or the pharmacy.

## Suggestions for improvements

The respondents made several suggestions on how information dissemination could be improved, primarily emphasising the need for information in more languages than Swedish. During times of less strict regulations, information could be disseminated at outdoor meetings. Some participants also expressed a sense of trust in health centres or pharmacies, and that these health institutions could have an expanded role in providing the local community with information. There were also suggestions that information regarding responsibilities and how to conduct safety at work could be sent by post to all people who were registered as running businesses. Since some immigrants, especially older people, have difficulties reading and writing, some also thought it would be

beneficial to produce audio-based information, which could be disseminated on government websites.

> Taxi driver: My grandfather will not understand printed information if you give him this leaflet, information should be in audio format and easy to understand.

> Property caretaker: Authorities who know that I run a business should have contacted me, they should have sent me a letter or texted me.

### Foreign-born persons' position
Some aspects were primarily tied to having an immigrant background and these were sorted into the following subthemes: 'exploitation of a vulnerable position' and 'protective directives not adapted for persons outside the majority'.

#### Exploitation of a vulnerable position
One of the most salient observations from the union representatives was difficulties in improving protection in cases where employment was a prerequisite to upholding a Swedish residency permit. As told by some of the union representatives, there are several Swedish vocational training programmes designed for new immigrants and therefore a high proportion of foreign-born persons without permanent residency in some occupations, which sometimes could be taken advantage of. Some employers also appeared to be aware that many immigrants need to financially support their relatives, which also might be abused, as speaking up against poor working conditions could result in fewer working hours or being assigned the worst shifts. Such aspects were also a reason for not joining the union since the union does not accept unregulated overtime.

> Union representative for bus drivers: We often notice that newly arrived persons attend SFI (comment: Swedish For Immigrants) training programs and the vocational bus training program. You are then completely dependent on your boss to get a residency permit. They would rather drive and get the wrong salary to get a residence permit. Because of this, I have chosen representatives against these employers who actually choose to exploit these people. This is almost modern slavery, "you do as I say, or you go home" (comment: she takes extra care to choose union representatives to combat these issues).

> Union representative taxi driver: Many of them then want to work a lot, they want money. Many send money home, many come from vulnerable (residential) areas so their children should get the same things as other children therefore they are at a disadvantage. Our driving and rest time rules mean that they are quite limited in how much they can work. If you're in the union, you can't cheat.

> Interviewer: Do the employers know about this?

> Union representative taxi driver: Oh yes.

> Interviewer: Do you think this is the case in your industry?

> Union representative taxi driver: Yes absolutely.

#### Protective directives not adapted for persons outside the majority of society
The participants reported that many of the safety recommendations disseminated by the Public Health Agency of Sweden, such as working from home or avoiding public transport, could not be adapted to the participants' life situation. One new theme that arose, for which there was no adapted information, was large social gatherings with persons from the same country, often privately arranged at cafés or other local venues. Other examples were large gatherings in cramped local food stores, mostly owned by other immigrants.

> Pizza baker: We have a cafe where you gather after work… How many were there last Friday or Saturday…I don't think there were more than 40 people (comment: the interview was conducted when eight persons were the maximum number of allowed people at social gatherings).

> Pizza baker: The (local) grocery store, that's it, 1000 people who come to the grocery store, move all the time, (touch) all the stuff, the bags, so there's a risk in the grocery store.

One aspect of interest in this study was overcrowding, but none of the interviewed persons could be defined as living in overcrowded conditions. Some of the union representatives mentioned that their members lived in overcrowded housing, but since none of the respondents themselves lived like this, we could not ask for more detailed information. Only two workers and one union representative reported limited finances as an obstacle for sick leave. Otherwise, most of the respondents reasoned that one's health was more important than money or that they were financially secure enough to go on sick leave if they needed it.

### DISCUSSION
This explorative interview study resulted in four overarching themes. The first concerned virus exposure at work, mainly through large numbers of social interactions and a lack of safety culture. The second theme related to working in a society-deemed low-status occupation where the participants' work and they as workers were not adequately appreciated or protected, despite carrying out important tasks for societal function. The third theme revolved around a lack of health literacy which meant that participants had limited access to essential crisis information, and instead turned to other information channels, through which a lot of misinformation was spread, causing confusion and the use of maladapted protective behaviour. Lastly, there were risk aspects tied to being foreign-born, most notably a precarious position

in the labour market and the need to keep an employment to maintain residency in Sweden, which could lead to exploitation and difficulties in speaking up against hazardous working conditions. There was also a lack of directives that were adapted to persons outside the majority of society.

## Virus exposure at work

Concerning virus exposure at work, the media and public health organisations have mainly focused on the health and social care sector, but other industries and large occupational groups also experienced increased risks of virus exposure at work.[9–12] International studies have consistently found that working in a low-status occupation is associated with an increased risk of COVID-19. In this study, we took a closer look at five Swedish occupational groups.[11 12] Weighing in hard-to-avoid sources of potential infection and poor organisational safety culture, cleaners, taxi, bus and tram drivers seemed notably vulnerable to many and physically close social contacts, which may explain why infection and mortality remained high among bus, tram and taxi drivers throughout the pandemic.[15 16] The many social contacts could also contribute to an increased spread of the disease to other people, some of whom could be considered at-risk groups for developing severe COVID-19.

A few studies with taxi and public transport drivers have presented similar findings,[28] while little is known about cleaners, property caretakers and pizza bakers, with the latter perhaps being best comparable with fast-food workers from an international perspective. An American study among fast-food workers with data collected from both questionnaires (n=417) and interviews (n=15)[18] illustrated several occupational hazards. These were partly linked to employers' inadequate investment and interest in safety culture, such as not providing staff with facemasks. Forty per cent of workers needed to buy their own facemasks, and out of those, around 10% of respondents were unable to afford them. Many coworkers or customers did not wear facemasks, with frequent incidents of threats and verbal abuse from customers when the staff tried to make them adhere to safety protocols.[17] In contrast, the pizza bakers in our study were mostly self-employed and appeared to be very able to create protective routines that were respected by staff and customers.

## Aspects of low status and undervalued work

Amid the pandemic, the inequities between different groups of workers became apparent, as workers in higher-status jobs could protect themselves, for example, by working from home, while those in already precarious jobs needed to work on-site, exposing themselves to the risk of infection. In most countries, including Sweden, healthcare workers were hailed as heroes during the pandemic. However, the workers in our study, who also provide important functions for the upholding of society, did not receive the same widespread acclaim. Bus, tram and taxi drivers sustained the transport infrastructure

and cleaners had an enhanced role since increased cleaning was regarded as an important measure to prevent the spread of COVID-19. One qualitative study from the UK,[18] conducted among bus drivers and transport workers, found similar themes as this study, primarily a sense of being unprotected at work, late or no introduction of protective equipment, as well as comparing themselves with other essential workers but without the same level of appreciation.

The definition of a low-status job is often related to a lack of longer formal training requirements or how work tasks are valued. Our investigated occupations can, by this definition, be viewed as low-status jobs. As such, one respondent reported that the employer calculated the financial loss for protective interventions and weighted this against the safety of their staff. The distance between the workers, decision-makers and employers also resulted in frustration since the people in authority often lacked knowledge of their work conditions and how to create a protective work environment.

## Lack of access to information

Another finding was the double-sided lack of health literacy from both the receiver's and the sender's perspectives. When COVID-19 was declared a full-scale pandemic, it dominated the news flow worldwide and awareness of the disease was high, but crucial information on what constituted a risk for infection and protective measures did not reach everyone equally. A European Union (EU)-funded consortium (COVINFORM) concluded that morbidity in COVID-19 partly depended on 'vulnerability in relation to communication', meaning lack of access to, or poorly adapted, information.[29] A Swedish study conducted before the pandemic illustrated that in geographical areas marked by low socioeconomic status and with a large proportion of first-generation and second-generation immigrants, there is a higher use of social media, foreign news media or social networks as the main source of information, especially among people with limited Swedish language proficiency.[30] This difficulty in accessing important societal information is referred to as 'the knowledge gap' and has been observed in several countries, in which foreign-born persons have more difficulties accessing crisis information since they lack social connections with knowledge about the new country or that they distrust authorities.[23] Health literacy tends to focus on the skills of the recipient, but it is indeed a two-way process in which the disseminator, in this context, employers and the Swedish health authorities, have a responsibility for disseminating information in a manner that is both received, understood and adapted to the communication patterns of the receiver. This is not only the ethically correct approach but also has a bearing on fundamental human rights in being given the possibility to protect one's health.

## Foreign-born persons' position

International studies have nearly unanimously concluded that the risk of contracting COVID-19 is higher in

immigrants in low-status jobs, a group of workers that even before the pandemic were at larger risk of work-related ill-health. A review study by the Swedish Work Environment Authority showed that individuals with an immigrant background more often worked in stressful jobs that required short formal training and in these occupations were assigned the worst tasks or received the least safety training.[31] Unfortunately, people from these social groups tend to not participate in scientific studies, partly due to a lack of trust, but also since participation often requires language proficiency to, for example, fill in questionnaires. Studies based on self-reported data are then not only scarce but also often biased, reflecting the experiences of those who are more integrated into Swedish society.

One of the most alarming findings in this study was the occurrence of exploitation, as some employers appeared to abuse the fact that many immigrants are dependent on their employment to keep their Swedish residency permit. These findings correspond with studies from other countries in which immigrants, especially those newly arrived, more often work in precarious work situations due to problems entering the labour market and the dependency on their jobs to legally remain in the new country.[32 33] Furthermore, most of the occupations investigated in this study require short or no formal training and no need for high language skills, which likely means that there is a large pool of available workers and that it is easy to replace people who are perceived as difficult or demanding. Several union representatives reported that speaking up could lead to punishments such as being assigned the 'graveyard shifts' or getting too few hours to fill a work week. One qualitative interview study among foreign-born cleaners in Sweden[34] illustrated a high competition between cleaning companies, but also many available workers which contributed to difficulties in receiving enough shifts to fulfil 100% employment. Due to these factors, the best shifts and a full-time schedule were often given to the workers who did not complain and who were on good terms with the foremen and employers. This made it difficult, if not impossible, to demand improved work conditions. If similar tendencies exist in the occupations selected for our study, it further illustrates how the workers are exposed to large risks at work, as well as being powerless over the conditions that expose them to a potentially lethal disease. Little is still known about long-term ill-health from COVID-19, but this already vulnerable group may also face higher risks of future reduced workability from post-COVID-19.[35] In light of such disparities, there has been a debate as to whether health organisations such as the WHO have a responsibility to enforce directives to protect workers.[36]

### Other considerations

This study had an explorative approach and used a set of questions with the possibility of follow-up questions to investigate the study's aims. Because of the qualitative design, external validity and generalisability cannot

be tested. The participants commonly described similar occurrences of risk situations for exposure to SARS-CoV-2, and the stability of these responses indicates that the results have transferability to similar job situations. This was further re-emphasised by the interviews with union representatives who have an overview of each occupation from representing large numbers of unionised members.

Our findings concur with issues of lack of information to all members of society, brought forth by the COVINFORM which overviews all the EU member states.[29] One of the consortium's members, the Birmingham Local Outbreak Engagement Board, has introduced several policies and trials of accessible COVID-19 information to the city of Birmingham (UK), home to 187 different nationalities.[37] Foremost, establishing multiple information channels, together with local organisations, community centres, faith settings, schools and faith institutions. This stands in contrast to a similar try-out in Norway, aimed at the Somali community, which was less successful due to not using already established communication channels.[38] As many shortcomings in the Swedish strategies can be traced to a communication distance between policymakers and some groups in societies, future policies should aim at giving people outside the majority of society the opportunity for influence, by being provided with communication channels to those responsible for compiling and disseminating information. In this way, knowledge of communication channels, special needs and unknown risks can increase, and public health interventions can be designed with better precision.

Finally, there is a risk that a study on this particular topic may support a discourse of blame and stigmatisation. Even before the COVID-19 pandemic, the public debate concerning certain demographic groups and disadvantaged geographical areas was characterised by a negative and one-sided approach. Such problematic images are at risk of being exacerbated, as certain urban areas or ethnic groups displayed a higher occurrence of COVID-19, at the expense of understanding the underlying causes. It is therefore important that future studies acknowledge the existing structural inequalities to avoid that groups with limited resources to govern their life situation being blamed for increasing the spread of a disease.

### Limitations

Based on the language proficiency of the interviewers and the respondents, we only conducted interviews in Swedish, Arabic and Somali. Four interviews were conducted in a language (Swedish) that was neither the respondent's nor the interviewer's native language, which likely restricted a mutual deeper understanding. This was possibly restricted further by that some interviews were conducted by telephone or Zoom, creating a larger communication distance. It is also likely that those who volunteered were a selection of more well-established people and with a higher level of trust in Swedish society and that we failed to recruit those most at risk. Future studies should conduct data collection in every

participant's native language and preferably by someone who also is knowledgeable about the participants' culture. There should also be more efforts to facilitate trust so that people who can be defined as marginalised feel comfortable to participate.

Almost all the participants were men since most of the selected occupations are male dominated. Nevertheless, foreign-born women more often report poorer working conditions or leaving the labour market prematurely, compared with both foreign-born men and Swedish-born women.[27] Because majority of the participants were male, aspects of dual vulnerability based on gender and immigrant status could not be elaborated. The three cleaners all worked at hospitals, while the union representatives all worked as general cleaners. Also, since none of the participants could be considered as living in an overcrowded household, we could not investigate this aspect.

## Strengths

This study has several strengths. First and foremost, that we could recruit participants from sociodemographic groups who previously have been identified as most at risk of COVID-19 infection, but who usually do not participate in scientific studies.[22] The lack of knowledge of the circumstances in such groups leads to a risk that public health interventions are not designed for those who need them the most. The successful recruitment was made possible because of the contracted health guides who, through their work, had a broad trust-based social network and were able to conduct several of the interviews in the participant's native language. Since relatively little is known about the work hazards and exposure risks of these groups in their private life, it is also an advantage that we have used qualitative methodology, which is an effective tool to explore new perspectives, since it makes it possible to capture nuances and delve into new themes with follow-up questions. We also interviewed both workers and union representatives. Through the latter, we could receive a broader perspective and gain information on negotiations with employers. Additionally, all union representatives were interviewed in February and March 2021, which meant they could contribute knowledge from different phases of the pandemic.

## CONCLUSIONS

In an increasingly globalised world, pandemics are likely to reappear and when widespread diseases enter, all members of society should be protected equally. This interview study highlights several potential risks of exposure to COVID-19 and the lack of possible protection in jobs outside the health and medical care sector dominated by foreign-born workers, as well as problems with the lack of adapted information. Our results are transferable to similar contexts and bring forth aspects that can be further explored in quantitative studies and public health interventions.

**Author affiliations**
<sup>1</sup>Occupational and Environmental Medicine, School of Public Health and Community Medicine, Sahlgrenska Academy, University of Gothenburg, Gothenburg, Sweden
<sup>2</sup>Angered Hospital, Hospitals West, Region Västra Götaland, Gothenburg, Sweden
<sup>3</sup>Occupational and Environmental Medicine, Sahlgrenska University Hospital, Region Västra Götaland, Gothenburg, Sweden
<sup>4</sup>Department of Molecular and Clinical Medicine, Sahlgrenska Academy, University of Gothenburg, Gothenburg, Sweden

**Contributors** MS, MM, KJ and AR conceived the study and were part of the overall conception and design of the study. MS coordinated the data collection and drafted the manuscript. MM and JS contributed to the data collection. All authors participated in the analysis of the data and critical revision of the manuscript. All authors have read and approved the final manuscript. MS is the guarantor of this study.

**Funding** This study was funded by the Swedish Research Council for Health, Working Life and Welfare (grant number: 2021-00326) and by the Swedish Research Council (grant number: 2021-06525).

**Competing interests** None declared.

**Patient and public involvement** Patients and/or the public were not involved in the design, or conduct, or reporting, or dissemination plans of this research.

**Patient consent for publication** Obtained.

**Ethics approval** This study involves human participants and was approved by the Swedish Ethics Review Authority (2021-00177). Participants gave informed consent to participate in the study before taking part.

**Provenance and peer review** Not commissioned; externally peer reviewed.

**Data availability statement** Data are available upon reasonable request. Data are not publicly available. Anonymised data are available upon reasonable request from the first author and pending fulfilment of legal requirements.

**ORCID iD**
Mia Söderberg http://orcid.org/0000-0003-0662-6541

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
