## [Reviewer comments · BMJ Open]

ARTICLE DETAILS

TITLE (PROVISIONAL)	Undervalued essential work and lacking health literacy as determinants of COVID-19 infection risks - a qualitative interview study among foreign-born workers in Sweden
AUTHORS	Söderberg, Mia; Magnusson, Maria; Swaid, Juhaina; Jakobsson, Kristina; Rosengren, Annika

VERSION 1 – REVIEW

REVIEWER	Tran, Nu Quy Linh Griffith University, School of Medicine and Dentistry
REVIEW RETURNED	15-Jan-2023

GENERAL COMMENTS	4. Are the methods described sufficiently to allow the study to be repeated? No Researchers did not sufficiently describe the recruitment method. Do not know how many people want to participate and how many meet the criteria. Why did the study end with 15 workers and 6 union representatives? 10. Are the results presented clearly? No - Some analyses lack quotes to support their conclusions - Reorganising and subdividing some themes is necessary to clarify the findings Please see the attached file for detailed comments 12. Are the limitations discussed adequately? No What were your efforts to minimise the limitations of the research? What are your suggestions to overcome them in the future? The research topic is quite interesting and has practical implications in the Public Health field. However, it has a lot of limitations, as follows: 1. Title Q1. The title is not clear, “interview study” is too general. Also, you should mention the setting (Sweden) 2. Background Q2. You should have 1 paragraph to introduce about covid-19 pandemic in Sweden. For example, severity, main waves, key prevention/response strategies, and especially the covid-19 vaccination cover rate right before conducting this study. Q3. You should identify knowledge gaps in existing evidence and the rationale of this study 3. Method Q4. What was the theory/framework used in this study to help design a research question, interpret the data, and explain of findings?
--

	Q5. The participants were recruited through social networks, which is a huge network. Why did only 15 workers and 6 union representatives participate? How many people responded and how many met the criteria? Q6. Line 20-21: What does “no patient involved” mean? Were all participants free of Covid-19 infection or did they not suffer from illness at the study time? Q7. Line 24: 15 workers were conducted from March 2021–March 2022. Did they participate in a single interview or a follow-up interview? Are there any significant changes in covid-19-related medical, social and political issues from Mar 2021 to Mar 2022 in Sweden? 4. Result Q8. There was no information about participants’ characteristics such as age, gender, number of working years, the covid-19 infection history, covid-19 vaccination. Q9. 1st sub-theme: Hard-to-avoid contamination risks Should be more explanation of why "hard to avoid". For example, a lack of occupational standards/policy/practices to prevent the disease, hard to apply prevention measures, or do not want to apply these methods. Need more explanation of the context Q10. 2nd sub-theme: lack of safety culture Line 43-44: “We asked why we are cleaning there - it's a risk. He (comment: their manager) says you must do it, there is no risk.” Did you double-check this information or ask more about it? why did a manager in a hospital say "there is no risk" of transmission in the emergency department? any misunderstanding or recall bias? 2.1 Fulfilling an unappreciated societal role Q11. Need more participants' quotes to confirm your statement 3.1 Health literacy Q12. The general statement is not clear Should be divided into sub-themes, for example, language barriers (Did they not understand or can not access the information not clear and consistent, or too much fake news?), the belief in the authority, etc 3.2 Exploitation Q13. Line 41-46: need participants’ quotes to support your statement 3.3. Protective directives not adapted for persons outside the majority Q14. Line 49-56: Were these the researcher's views or that of the participant? Do any direct quotes to support these statements?
--	--

REVIEWER	Yu, Jiao University of Minnesota
REVIEW RETURNED	28-Feb-2023

GENERAL COMMENTS	This study employs a qualitative research design and convenient sampling strategy to address the important question of what aspects of work and living conditions determine COVID-19 infection risks among foreign-born workers in non-health care occupations. It was generally well written, with adequate methods for investigating the research questions. However, there are some issues that could be addressed to strengthen the contributions of this paper. The paper would benefit from an explicit use of theory. It seems there is no theoretical framework that guides this study. Other than the prevalence of COVID-related outcomes, I also suggest the
---

	authors provide a literature review of work conditions, working autonomy, stress processes, and immigration status that relate to individual (health) outcomes. I suggest the authors provide more detailed information on the data collection method. For example, how the interviewers were trained to ensure the validity of data collection. Where were the interviews held? For the analysis, what procedures were employed to ensure the reliability and validity of the analytical results? Were all transcripts read and labeled by two researchers? When there was a disagreement between the two coders, was there a third person who could reevaluate the themes? Some description of the triangulation is needed. I appreciate the authors' presenting relevant transcripts to support their findings. I propose removing some redundancy. For example, I am aware there should be some occupation differences in the participants' working experiences during the COVID-19 pandemic. I suggest presenting several summary findings across these occupations rather than organizing the results based on occupation categories. I also wonder, because both workers and union members participated in this study, what insights the union members may have that we may not be able to gain from the workers. In addition, the strength of this study is the inclusion of the hard-to-reach sociodemographic groups. The author may highlight their findings based on this group that have not been revealed or thoroughly investigated in previous work, for example, what are those structural constraints these workers faced during the pandemic. Relevant public policy implications should be provided and discussed. What are the possible policy recommendations based on the current findings?
--	---

VERSION 1 – AUTHOR RESPONSE

Reviewer: 1

Dr. Nu Quy Linh Tran, Griffith University, Centre for Disease Control and Prevention, Ha Tinh province, Vietnam Comments to the Author

Comment: Below are responses to Reviewer 1 comments that were submitted in the BMJ open submission system and the reviewer's attached document with additional comments.

1. Title

Q1. The title is not clear, "interview study" is too general. Also, you should mention the setting (Sweden).

Response: We have changed the title accordingly.

2. Background

Q2. You should have 1 paragraph to introduce about covid-19 pandemic in Sweden. For example, severity, main waves, key prevention/response strategies, and especially the covid-19 vaccination cover rate right before conducting this study.

Response: We have added a paragraph in the introduction on the COVID public health strategies in Sweden.

Q3. You should identify knowledge gaps in existing evidence and the rationale of this study

Response: We have provided findings from existing studies and the lack of knowledge of risks for COVID-19 infection among foreign-born workers in sectors outside the health and social care sector in the introduction.

3. Method

Q4. What was the theory/framework used in this study to help design a research question, interpret the data, and explain of findings?

Response: Qualitative studies sometimes are used to analyse a theory through, for example qualitative deductive method, but it is common to use an explorative approach to gain new or in-depth knowledge into partly known or unknown issues. The current study explored aspects of a workday or living conditions that could constitute risks for COVID, rather than testing a theory. We have revised the introduction to put more emphasize into what is known, what can be expected based on the existing literature and what knowledge gaps this study may contribute with.

4. Are the methods described sufficiently to allow the study to be repeated? No Researchers did not sufficiently describe the recruitment method. Do not know how many people want to participate and how many meet the criteria. Why did the study end with 15 workers and 6 union representatives?

Response: We have revised the method section to clarify the recruitment method. All individuals that fulfilled the inclusion criteria and wanted to participate was included. The recruitment ended when data was considered to be saturated, meaning that there is no new information with each new interview and each new interview mainly consist of repetitive themes. This is a common criterion to end recruitment in qualitative studies.

Q5. The participants were recruited through social networks, which is a huge network. Why did only 15 workers and 6 union representatives participate? How many people responded and how many met the criteria?

Response: Although the social networks of the health guides, who were the main recruiters, are large we aimed to recruit foreign-born persons (second generation immigrants were not considered) who worked as pizza bakers, taxi drivers, bus and tram drivers, cleaners and property caretakers, which narrowed the pool of potential participants. Recruitment continued until data was saturated, meaning that the content became repetitive and no new information emerges. The number of participants is not as important in qualitative studies, as in quantitative studies since statistical interference and generalizability is not part of the method. Rather the empathises is to extract new information, or in-depth nuances of known information than can be explored in future quantitative studies. The results are discussed in terms of transferability in the discussion section.

Q6. Line 20-21: What does “no patient involved” mean? Were all participants free of Covid-19 infection or did they not suffer from illness at the study time?

Response: The sub-headline “Patient and public involvement” is a standard headline required by BMJ Open. The statement “No patients were involved in this study” refers to that participants were not recruited from a clinical setting. We have clarified this phrasing.

Q7. Line 24: 15 workers were conducted from March 2021–March 2022. Did they participate in a single interview or a follow-up interview? Are there any significant changes in covid-19-related medical, social and political issues from Mar 2021 to Mar 2022 in Sweden?

Response: All the participants were interviewed once. We have added a paragraph of information of COVID related public health strategies for Sweden.

4. Result

Q8. There was no information about participants' characteristics such as age, gender, number of working years, the covid-19 infection history, covid-19 vaccination.

Response: We have added this information and also inserted some of the information from appendix I to the main text in the results. As several of the participants were interviewed during a timepoint when they were not included in the Swedish national vaccination programme our interview guide did not include such questions.

Q9. 1st sub-theme: Hard-to-avoid contamination risks

Should be more explanation of why "hard to avoid". For example, a lack of occupational standards/policy/practices to prevent the disease, hard to apply prevention measures, or do not want to apply these methods. Need more explanation of the context.

Response: We have revised this section. However, we did not obtain any information on occupational standards/policies/practices from the workers and according to the union workers such aspects differed between employers, which is mentioned under the heading "lack of safety culture" and "Occupational status influences workers' value and safety".

10. Are the results presented clearly? No

- Some analyses lack quotes to support their conclusions
- Reorganising and subdividing some themes is necessary to clarify the findings.

Please see the attached file for detailed comments

Response: We have revised accordingly.

Q10. 2nd sub-theme: lack of safety culture Line 43-44: "We asked why we are cleaning there - it's a risk. He (comment: their manager) says you must do it, there is no risk."

Did you double-check this information or ask more about it? Why did a manager in a hospital say "there is no risk" of transmission in the emergency department? Any misunderstanding or recall bias?

Response: We have reported the findings according to the collected data, i.e., how the participants answered our questions. The manager refers to the cleaning manager, not the health care manager. We did talk to other cleaners and hospital staff "off the record" and from these reports it seemed like cleaners at some hospitals received little to no protective equipment or safety routines in the early phases of the pandemic. But since this information is not part of this study data collection such information is not included in the results.

2.1 Fulfilling an unappreciated societal role

Q11. Need more participants' quotes to confirm your statement

Response: We have added quotes.

3.1 Health literacy

12. Are the limitations discussed adequately? No. What were your efforts to minimise the limitations of the research? What are your suggestions to overcome them in the future?

Response: we have added such aspects to the discussion.

Q12. The general statement is not clear. Should be divided into sub-themes, for example, language barriers (Did they not understand or cannot access the information not clear and consistent, or too much fake news?), the belief in the authority, etc

Response: We have added lack of information as a new main theme and re-organized some of the sub-themes.

3.2 Exploitation

Q13. Line 41-46: need participants' quotes to support your statement

Response: We have added more quotes to the existing ones.

3.3. Protective directives not adapted for persons outside the majority

Q14. Line 49-56: Were these the researcher's views or that of the participant? Do any direct quotes to support these statements?

Response: we have clarified the phrasings and added quotes to this sub-theme.

Reviewer: 2

Dr. Jiao Yu, University of Minnesota

Comments to the Author:

This study employs a qualitative research design and convenient sampling strategy to address the important question of what aspects of work and living conditions determine COVID-19 infection risks among foreign-born workers in non-health care occupations. It was generally well written, with adequate methods for investigating the research questions. However, there are some issues that could be addressed to strengthen the contributions of this paper.

The paper would benefit from an explicit use of theory. It seems there is no theoretical framework that guides this study.

Response: Although qualitative studies may be used to analyse a theory through for example qualitative deductive method, it is common to use an explorative approach in order to gain new or a more in-depth insight into partly known or unknown issues. Thus, this study explored aspects of a typical workday or living conditions that could constitute risks for COVID infected, rather than testing a theory. We have, however, revised the introduction to put more emphasize into what is known, what can be expected based on the existing literature and what knowledge gaps this study may contribute with.

Other than the prevalence of COVID-related outcomes, I also suggest the authors provide a literature review of work conditions, working autonomy, stress processes, and immigration status that relate to individual (health) outcomes.

Response: We have added some information on the requested topic to the discussion, but since the research focus is infection risk exposures rather than the general impact of work environment on health, we kept it brief.

I suggest the authors provide more detailed information on the data collection method. For example, how the interviewers were trained to ensure the validity of data collection. Where were the interviews held?

Response: We have added this information to the method section

For the analysis, what procedures were employed to ensure the reliability and validity of the analytical results?

Response: We have added such aspects in the discussion under the heading "other considerations".

Were all transcripts read and labelled by two researchers? When there was a disagreement between the two coders, was there a third person who could re-evaluate the themes? Some description of the triangulation is needed.

Response: We have further clarified the method section. Only the first and second author read all the transcripts. The third author and health guide JS that conducted five of the interviews re-read the transcripts of all the interviews she carried out (in Arabic). Both her and the other health guide Saido Mohamed that conducted four of the interviews (Somali), revised the themes and pointed out

information that might have been missed and if the themes corresponded to their impression during the interviews.

I appreciate the authors' presenting relevant transcripts to support their findings. I propose removing some redundancy. For example, I am aware there should be some occupation differences in the participants' working experiences during the COVID-19 pandemic. I suggest presenting several summary findings across these occupations rather than organizing the results based on occupation categories.

Response: We agree and have revised the results accordingly.

I also wonder, because both workers and union members participated in this study, what insights the union members may have that we may not be able to gain from the workers.

Response: We have added such reasoning to the discussion.

In addition, the strength of this study is the inclusion of the hard-to-reach sociodemographic groups. The author may highlight their findings based on this group that have not been revealed or thoroughly investigated in previous work, for example, what are those structural constraints these workers faced during the pandemic.

Response: We agree and have added this to the discussion.

Relevant public policy implications should be provided and discussed. What are the possible policy recommendations based on the current findings?

Response: Although we are grateful that the reviewer highlights the importance of the findings in this study, the number of participants is perhaps too small to discuss policy implication. To create a knowledge foundation for policy implications it would be beneficial to try and evaluate interventions in different populations or try replicate similar results as in this study in larger quantitative studies.

VERSION 2 – REVIEW

REVIEWER	Tran, Nu Quy Linh Griffith University, School of Medicine and Dentistry
REVIEW RETURNED	26-Sep-2023

GENERAL COMMENTS	After thorough review and careful consideration, I think the manuscript, in its present form, does not meet the publication standards. Key concerns include:  1. Lack of Novel Contribution: The determinants of COVID-19 infection risks are very complex and have been extensively studied. The manuscript does not significantly advance our understanding or present new insights. 2. Incomplete Literature Review: The background/literature review section is not adequately developed. Several pieces of information, particularly in the first and second paragraphs of the background section, are presented without proper citation. The manuscript fails to synthesize previous evidence effectively and does not provide a compelling rationale for the significance of the current study. 3. Methodological Concerns: The interview guide and interview questions did not adequately address the research aim. The qualitative method, using semi-structured interviews with 15 workers across diverse jobs (pizza bakers, property caretakers, taxi, tram or bus drivers, and cleaners) may not provide enough understanding. Due to the disparate nature of these jobs, which inherently present different risks of COVID-19 infection, the variance in responses may introduce complexities in drawing meaningful conclusions.
--

	Methodological Issues: the interview guide and questions were not sufficiently tailored to address the study's objective. The research employed semi-structured interviews with a relatively small sample of 15 workers and 6 union representatives engaged in various professions (pizza bakers, property caretakers, and individuals driving taxis, trams, or buses). For each distinct job, the representation was limited to merely two to three participants, which raises concerns regarding the sufficient information to synthesize the findings and draw representative conclusions. Also, each of these occupations carries distinct risks associated with COVID-19 infection, leading to a considerable degree of variability in the participants' responses. However, the manuscript has not inadequately presented this issue The data analysis was also found to be in its preliminary stages, primarily focused on simple coding, limiting the depth and reliability of the results. Additionally, the limitations associated with utilizing Zoom and telephone for conducting qualitative interviews were not adequately addressed or discussed in the study. 4. Also, the manuscript does not offer practical recommendations for future pandemic preparedness and adaptation. The manuscript needs to articulate clearly its contributions and how it fits with or expands upon existing literature on the topic.
--	--

REVIEWER	Yu, Jiao University of Minnesota
REVIEW RETURNED	11-Sep-2023

GENERAL COMMENTS	The paper has undergone noteworthy improvements and the authors have addressed most of my prior concerns. A minor point is that I would like to suggest a discussion on policy implications. Despite the small sample size, the study's results could still provide invaluable insights, and recommendations for policies could be immensely beneficial.
---

VERSION 2 – AUTHOR RESPONSE

Reviewer: 2

Dr. Jiao Yu, University of Minnesota

Comments to the Author:

The paper has undergone noteworthy improvements and the authors have addressed most of my prior concerns. A minor point is that I would like to suggest a discussion on policy implications. Despite the small sample size, the study's results could still provide invaluable insights, and recommendations for policies could be immensely beneficial.

Response: We appreciate the feedback and have added suggestions for policy changes on page 19.

Reviewer: 1

Dr. Nu Quy Linh Tran, Griffith University, Centre for Disease Control and Prevention, Ha Tinh province, Vietnam

Comments to the Author:

After thorough review and careful consideration, I think the manuscript, in its present form, does not meet the publication standards.

Key concerns include:

1. Lack of Novel Contribution: The determinants of COVID-19 infection risks are very complex and have been extensively studied. The manuscript does not significantly advance our understanding or present new insights.

Response: While several pathways for infection risks have been examined thoroughly, there are very few studies on occupational risks for COVID-19 virus exposure among foreign-born workers in non-healthcare and low-salary occupation, especially as reported by these workers themselves. One reason is that foreign-born workers more often decline participation in scientific studies. Response rates are even lower in foreign-born persons with short education. Additionally, in quantitative studies, questionnaires are too often not translated to their native languages, meaning that those with limited language skills, who are likely the most vulnerable, cannot participate or provide poor quality data since they do not fully understand the content of the questionnaires.

A novelty of this study is the use of a qualitative design – which has now been added to the introduction in order to clarify further why this study was done (page 5). Using a qualitative study design, we had the possibility to capture aspect that are particular to this hard-to-reach group and who among working-aged persons in Sweden displayed among the highest risks for infection and severe disease. We could also conduct the interviews in several of the participants' native language. Since this is research that can explore and provide more profound insights alongside quantitative research, it can be helpful in advancing our understanding of living conditions in the job categories that we approached, new information that can be further explored in future qualitative as well as quantitative.

2. Incomplete Literature Review: The background/literature review section is not adequately developed. Several pieces of information, particularly in the first and second paragraphs of the background section, are presented without proper citation. The manuscript fails to synthesize previous evidence effectively and does not provide a compelling rationale for the significance of the current study.

Response: We have added references to the first and second paragraph. Furthermore, we have presented the rationale for our study, based on evidence of higher infection risks in foreign-born persons and in specific occupations, yet self-reported infection risk situations in their work routine or daily life have been little explored, summed up on page 5. The lack of studies on risks in these occupations and among persons with an immigrant background, in particular among people with limited skills in the language in the new country, means that little is known about these peoples' work and life circumstances, making an exploratory approach suitable.

3. Methodological Concerns: The interview guide and interview questions did not adequately address the research aim. The qualitative method, using semi-structured interviews with 15 workers across diverse jobs (pizza bakers, property caretakers, taxi, tram or bus drivers, and cleaners) may not provide enough understanding. Due to the disparate nature of these jobs, which inherently present different risks of COVID-19 infection. the variance in responses may introduce complexities in drawing meaningful conclusions.

Response: The interview questions stated in the method section is not the full interview guide, but examples of questions. Furthermore, this is an exploratory study, in which it is common to have broad, general, and open questions since little is previously known. Then, the participants had the possibility to answer freely and elaborate around each question which then could be followed-up by further questions to extract as much new information as possible. With this method new topics and information, and in-depth knowledge or nuances of partly known topic can be captured.

In the first submitted version we included several findings of work conditions that were particular to each occupation. However, per suggestion from reviewer 2, we revised the manuscript to mostly focus on aspects that are similar among the participants, such as many close physical contacts. Still, we report some occupation specific findings, that we considered particularly noteworthy. For example, that bus drivers needed to facilitate ticket inspection or that hospital cleaner needed to clean wards where infected patients had been admitted.

Methodological Issues: the interview guide and questions were not sufficiently tailored to address the study's objective. The research employed semi-structured interviews with a relatively small sample of 15 workers and 6 union representatives engaged in various professions (pizza bakers, property caretakers, and individuals driving taxis, trams, or buses). For each distinct job, the representation was limited to merely two to three participants, which raises concerns regarding the sufficient information to synthesize the findings and draw representative conclusions. Also, each of these occupations carries distinct risks associated with COVID-19 infection, leading to a considerable degree of variability in the participants' responses. However, the manuscript has not inadequately presented this issue

Response: Since this is a qualitative study, statistical interference and generalizability cannot be evaluated, and is also not the point. Instead, qualitative studies are typically used to explore new topics, or to gain in-depth knowledge or nuances of known or partly known topics. The data collection is usually ended when, through the course of interviewing, the same themes coming out repeatedly and you stop finding new themes, ideas, opinions, or patterns. We would also like to add that qualitative research, such as the one we used for the present study, is rigorously designed and with protocols that are followed, just as in quantitative research. We added a few sentences in the introduction to further clarify this. Findings from the qualitative study then can be further explored in larger samples using a quantitative design. There were differences between the occupations but per suggestion by the previous round of reviewer comments we have focused on similarities between the occupations.

The data analysis was also found to be in its preliminary stages, primarily focused on simple coding, limiting the depth and reliability of the results. Additionally, the limitations associated with utilizing Zoom and telephone for conducting qualitative interviews were not adequately addressed or discussed in the study.

Response: One way of evaluating reliability in qualitative studies is to individually extract meaning bearing units and overarching themes and see if the different interpretations are consistent. In this case three of the authors read the transcripts and compared found themes. An additional person, a contracted health guide with deep contextual knowledge also contributed with input on the result. We have added the shortcomings of using Zoom and telephone to the to the limitations.

4. Also, the manuscript does not offer practical recommendations for future pandemic preparedness and adaptation. The manuscript needs to articulate clearly its contributions and how it fits with or expands upon existing literature on the topic.

Response: We have added recommendations on page 19. Contributions of this study are stated in conclusions on page 20.